# Oral Metronomic Maintenance Therapy Can Improve Survival in High-Risk Neuroblastoma Patients Not Treated with ASCT or Anti-GD2 Antibodies

**DOI:** 10.3390/cancers13143494

**Published:** 2021-07-13

**Authors:** Xiaofei Sun, Zijun Zhen, Ying Guo, Yuanhong Gao, Juan Wang, Yu Zhang, Jia Zhu, Suying Lu, Feifei Sun, Junting Huang, Ruiqing Cai, Yizhuo Zhang, Juncheng Liu, Zizheng Xiao, Sihui Zeng, Zhuowei Liu

**Affiliations:** 1State Key Laboratory of Oncology in South China, Guanghzou 510060, China; zhenzj@sysucc.org.cn (Z.Z.); guoying@sysucc.org.cn (Y.G.); gaoyh@sysucc.org.cn (Y.G.); wangjuan@sysucc.org.cn (J.W.); zhangy@sysucc.org.cn (Y.Z.); zhuj@sysucc.org.cn (J.Z.); lusy@sysucc.org.cn (S.L.); sunff@sysucc.org.cn (F.S.); huangjt@sysucc.org.cn (J.H.); cairq@sysucc.org.cn (R.C.); zhangyz@sysucc.org.cn (Y.Z.); xiaozzh@sysucc.org.cn (Z.X.); zengsh@sysucc.org.cn (S.Z.); 2Department of Pediatric Oncology, Sun Yat-sen University Cancer Center, Guanghzou 510060, China; 3Department of Clinical Research, Sun Yat-sen University Cancer Center, Guanghzou 510060, China; 4Department of Radiotherapy, Sun Yat-sen University Cancer Center, Guanghzou 510060, China; 5Department of Pathology, Sun Yat-sen University Cancer Center, Guanghzou 510060, China; 6Department of Pediatric Surgery, First Affiliated Hospital of Sun Yat-sen University, Guangzhou 510060, China; liujch@mail.sysu.edu.cn; 7Department of Nuclear Medicine, Sun Yat-sen University Cancer Center, Guanghzou 510060, China; 8Department of Radiology, Sun Yat-sen University Cancer Center, Guanghzou 510060, China; 9Department of Urology Surgery, Sun Yat-sen University Cancer Center, Guanghzou 510060, China

**Keywords:** neuroblastoma, metronomic chemotherapy, maintenance therapy, high-risk

## Abstract

**Simple Summary:**

Low-dose metronomic chemotherapy has anti-angiogenic activity and inhibits tumor growth. Therefore, we investigated the benefits of low-dose metronomic maintenance therapy (MT) in high-risk neuroblastoma (NB) patients who are unable to undergo autologous stem cell transplantation (ASCT) or anti-GD2 antibody therapy. A total of 217 high-risk NB patients were enrolled. One hundred and eighty-five (85%) had a complete/very good partial remission/partial remission (CR/VGPR/PR) to treatment, of them, 167 patients with stage 4, that did or did not receive oral metronomic MT, 3 years of event-free survival (EFS) were 42.5% versus 29.4%, and overall survival (OS) was 71.1% versus 59.4%, respectively. Totally, 117 high-risk patients with oral metronomic MT had an EFS rate of 42.7%. The results were similar to those of ASCT from other studies. The toxicities of metronomic MT were lower. Our study showed that oral metronomic MT is an optimal option for high-risk NB patients without ASCT or anti-GD2 antibody therapy.

**Abstract:**

Despite aggressive treatment, the prognosis of high-risk NB patients is still poor. This retrospective study investigated the benefits of metronomic maintenance treatment (MT) in high-risk NB patients without ASCT or GD2 antibody therapy. Patients aged ≤ 21 years with newly diagnosed high-risk NB were included. Patients with complete/very good partial remission (CR/VGPR/PR) to conventional treatment received, or not, oral metronomic MT for 1 year. Two hundred and seventeen high-risk NB patients were enrolled. One hundred and eighty-five (85%) had a CR/VGPR/PR to conventional treatment, of the patients with stage 4, 106 receiving and 61 not receiving oral metronomic MT, and the 3-year event-free survival (EFS) rate was 42.5 ± 5.1% and 29.6 ± 6%, respectively (*p* = 0.017), and overall survival (OS) rate was 71.1 ± 4.7% and 59.4 ± 6.4%, respectively (*p* = 0.022). A total of 117 high-risk patients with oral metronomic MT had EFS rate of 42.7 ± 4.8%. The toxicity of MT was mild. For high-risk NB patients without ASCT or anti-GD2 antibody therapy, stage 4, MYCN amplication and patients with stage 4 not receiving oral metronomic MT after CR/VGPR/PR were independent adverse prognostic factors. Oral metronomic MT can improve survival in high-risk NB patients in CR/VGPR/PR without ASCT or anti-GD2 antibodies therapy.

## 1. Introduction

Neuroblastoma (NB) is the most common non-central nervous system pediatric malignant solid tumor. Children with low- or intermediate-risk NB have an excellent prognosis. However, high-risk NB patients have a poor prognosis, and their 3-year event-free survival (EFS) rate is about 40% among patients who receive induction chemotherapy, surgery, single high-dose chemotherapy with autologous stem cell transplantation (ASCT), and radiotherapy plus cis-retinoic acid maintenance therapy (MT) [1,2,3,4,5,6]. The survival rate of high-risk NB has increased to 65–75% when ASCT is combined with anti-GD2 antibody in the modern immunotherapy era [7,8,9,10]. Therefore, anti-GD2 antibodies immunotherapy after ASCT has been considered as the standard of care for high-risk NB disease.

ASCT is one of the consolidation therapies options for high-risk NB patients. Three large randomized controlled studies have shown an improvement in the 3-year EFS after ASCT (31% to 47%) compared to that after 4 courses of conventional chemotherapy or oral chemotherapy or no further treatment (22% to 31%) [1,2,3]. Single ASCT may increase EFS by about 10% in high-risk NB patients. High-risk NB patients can benefit from ASCT in terms of 3 to 5 years of EFS, while there is no OS benefit [1,2,3,4]. But long-term outcomes of the GPOH NB97 trial showed 10 years of OS were in favor of the ASCT groups [5], yet the other randomized studies showed that the difference in 10 years of OS was still no statistically significant [6]. However, few families in China have access to or can afford expensive anti-GD2 antibody therapy, and about 70% of high-risk NB patients who have achieved complete/very good partial remission (CR/VGPR) could not undergo ASCT for various reasons, including cost, poor physical tolerance, insufficient supportive care, or family reluctance. The high recurrence rate after ASCT is one of the main reasons that many families do not choose ASCT. For these patients, the survival rate is only 20–30%, thus the need to develop other treatments to delay recurrence and prolong survival. The continuous administration of oral low-dose metronomic MT is one of the options worthy of exploring.

A number of studies have shown that low-dose metronomic chemotherapy has anti-angiogenic activity and inhibit tumor growth, while stimulating the immune system with minimal toxicity, and has a potential to develop a tumor maintenance therapy [11,12,13,14]. The continuous administration of low-dose metronomic MT has been successfully used for decades in pediatric patients with acute lymphoblastic leukemia (ALL) [15]. The HD CWS-96 trial compared children with metastatic soft tissue sarcoma undergoing oral MT and high-dose chemotherapy and found that an oral MT was feasible and tolerable, with surprisingly a better outcome than high-dose salvage chemotherapy (5-year OS 52% versus 27%) [16]. Recent results from a multicenter, open-label, randomized, phase 3 trial from the European Pediatric Soft Tissue Sarcoma Study Group for patients with high-risk rhabdomyosarcoma showed that vinorelbine and continuous low-dose cyclophosphamide as MT improved the survival of patients with high-risk rhabdomyosarcoma. This approach will be the new standard of care for patients with high-risk rhabdomyosarcoma in the future [17]. This study has put low-dose metronomic MT in high-risk solid tumors back into the spotlight. However, there are few reports about low-dose metronomic MT in high-risk NB patients.

Since 2013, we have administered oral low-dose metronomic MT to high-risk NB patients with CR/VGPR/PR who are not treated with ASCT and/or anti-GD2 antibodies. The purpose of study was to determine whether those patients can benefit from oral metronomic MT. The prognostic factors among high-risk NB in this unique cohort Chinese children were also analyzed.

## 2. Materials and Methods

### 2.1. Patients

Children younger than 21 years with previously untreated high-risk NB who were treated at the Sun Yat-sen University Cancer Center (SYSUCC) between January 2013 and December 2018 were eligible. The high-risk NB patients who did not receive ASCT and/or anti-GD2 immunotherapy were enrolled to study. High-risk NB was defined as follows: (1) INSS stage 4 disease and age ≥ 18 months; (2) INSS stage 2, 3, and 4S disease and *MYCN* amplification; and (3) INSS stage 3 disease and age > 18 months with unfavorable pathology without *MYCN* amplification [18]. Informed consent was obtained from patients and their parents, and complete treatment and follow-up data were available. This study was approved by the Ethics Committee of SYSUCC.

The extent of tumor spread was evaluated by CT and/or MRI, bilateral bone marrow examination, ^99^Tc bone scan, or ^18^F-fluorodeoxy-D-glucose positron emission tomography/CT (MIBG scan was not available in China). Clinical stage was determined based on the International Neuroblastoma Staging System [19]. The Pathology Classification system was used for pathologic classification of tumors [18]. MYCN amplification was examined using fluorescence in situ hybridization.

### 2.2. Treatment Protocol

Patients received eight cycles of induction chemotherapy, surgery, local radiotherapy, followed by MT when they could not receive ASCT and/or anti-GD2 antibodies. For induction chemotherapy, the CAV (cyclophosphamide, pirarubicin, vincristine) and VIP (etoposide, ifosfamide, cisplatin) regimens were administered alternatively at 3-week intervals. Surgery was usually performed after four to six cycles of chemotherapy. If the tumor was deemed unresectable, patients were treated with another two cycles of CAV/VIP or changed to second-line chemotherapy consisting of vincristine, irinotecan, and temozolomide (VIT). Local radiotherapy 25–30 Gy was administered to all patients after surgery. Patients who achieved CR/VGPR/PR after comprehensive treatment were treated with low dose oral metronomic anti-angiogenic drugs (cyclophosphamide, vinorelbine, etoposide and/or topotecan, and celecoxib) as MT for 1 year (Table 1). The starting time of oral metronomic MT was that peripheral white blood cells reached 3 × 10^9^/L after end of comprehensive therapy. Extent of disease evaluation was performed every 3 months during MT.

### 2.3. Evaluation of Response and Toxicity 

Treatment response was evaluated every two cycles according to the international response standard for NB [19]. CR was defined as the absence of tumors with normal catecholamine levels. VGPR was defined as a 90% ~ 99% reduction in primary tumors, elimination of all measurable metastatic disease, normal catecholamine levels, with or without residual ^99^Tc bone changes. PR was defined as a reduction of > 50% in primary and metastatic tumors. Stable disease (SD) was defined as the absence of new lesions or an increase of < 25% in existing lesions. Progressive disease (PD) was defined as any new lesion or any measurable lesions increasing by > 25%. Toxicity was assessed according to the Common Terminology Standard for Adverse Events version 4.03.

### 2.4. Statistical Analysis

Statistical analyses were performed using the SPSS, version 22 (IBM Corp, New York, NY, USA) and Stata software, version 15.1 (Stata Corp LLC TX, USA). Survival end points were described by their rate at specific time points with a 95% CI. Both EFS and OS were analyzed using the Kaplan-Meier method and compared using the log-rank test. Hazard ratios were calculated using univariable and multivariable Cox proportional hazards regression analyses. All statistical tests were two-sided, and a difference was considered significant when the *p*-value was < 0.05. Event-free survival was calculated as the time from enrollment to the first occurrence of relapse, progression, death from any cause, secondary cancer, or the time of the last contact if no event has occurred. Overall survival was calculated as the time from diagnosis to death or the last examination when patient remained alive. The last update on 31 October 2020 was used for this analysis.

## 3. Results

### 3.1. Patient Characteristics

A total of 217 patients newly diagnosed with high-risk NB were included in this study. The median age was 3.7 years (range: 0.5–21 years). Among them, one hundred and ninety-eight patients (91.2%) had stage 4 disease, and 172 (79.3%) had bone marrow/bone metastases. MYCN amplification was assessed in 160 tumors (73.7%), of which 46 (28.8%) were positive. The clinical characteristics of the patients are shown in Table 2.

### 3.2. Treatment Outcome

Among 217 patients with high-risk NB, 171 (78.8%) were treated with chemotherapy, surgery, and radiotherapy, 29 (13.4%) were treated with chemotherapy plus surgery, and 17 (7.8%) were treated with chemotherapy alone. After induction therapy, the extent of resection of the primary tumor was ≥ 90% in 164 (75.5%) patients. Fifty-nine patients achieved CR, 126 achieved VGPR/PR, 14 had SD, and 18 had PD to comprehensive therapies. The overall response rate was 85.2%. Among the 185 patients with CR/VGPR/PR, a total of 117 received oral metronomic MT, and 68 patients did not receive oral MT; of them, 28 cases received isotretinoin, and the remaining 40 cases received Chinese herbal medicine or other therapy, or no therapy. In 167 patients with stage 4, 106 received oral metronomic MT, and 61 did not receive oral metronomic MT (Figure 1).

A total of 126 patients experienced tumor recurrence or progression. Of these, 40 abandoned further therapy, and 86 patients received various salvage treatments, including chemotherapy, surgery, and radiotherapy. Anti-angiogenesis drugs, such as apatinib or arotinib, were administered alone or in combination with the salvage chemotherapy regimens.

### 3.3. Survival

The median follow-up was 41.3 months (range: 6.8–88.0 months). A total of 126 patients experienced tumor recurrence or progression, and 104 patients died. In the entire cohort, the 3-year EFS and OS rates were 36.3 ± 3.4% and 63.1 ± 3.4%, respectively. The EFS rates of patients with stage 4 disease was significantly lower than that of patients with stage 3/4S disease (33.8 ± 3.5% versus 62.2 ± 11.4%, *p =* 0.011). The EFS rates of patients with and without MYCN amplification were 28.6 ± 7.0% and 44.3 ± 4.8%, respectively (*p* = 0.038), and the corresponding OS rates were 48.6 ± 8.0% and 74 ± 4.3%, respectively (*p =* 0.008) (Table 3 and Figure 2).

The 3-years EFS rates of the 167 high-risk patients with stage 4 who received or did not receive oral metronomic MT after achieving CR/VGPR/PR (Cohort 1) were 42.5 ± 5.1% and 29.6 ± 6%, respectively (*p =* 0.017), and OS was 71.1 ± 4.7% and 59.4 ± 6.4%, respectively (*p* = 0.022). The patients with MYCN amplification had worse 3-year EFS and OS than patients without MYCN amplification (12.3% versus 46.1%, *p =* 0.019, and 37.8% versus 76.7%, *p* = 0.000) (Table 3 and Figure 2).

The 3-year EFS and OS rates of the 117 high-risk patients who received oral low dose metronomic MT after achieving CR/VGPR/PR (Cohort 2) were 42.7% ± 4.8% and 72.1% ± 4.5%, respectively. Further, among these patients, EFS and OS rates of those with MYCN amplification were worse than those without MYCN amplification (16.8% versus 53.9%, *p =* 0.026, and 56.3% versus 78.1%, *p =* 0.042) (Table 3 and Figure 2).

In univariate analysis, stage 4 disease, MYCN amplification and stage 4 without oral metronomic MT were significant adverse prognostic factors for EFS and OS. Multivariate analysis showed that the stage 4 disease (hazard ratio (HR) 0.246, *p =* 0.002), MYCN amplification (HR 0.462, *p =* 0.001), and stage 4 without oral metronomic MT (HR 0.559, *p =* 0.014) were predictive of an adverse EFS prognosis. For OS, the stage 4 disease (HR 0.147, *p =* 0.003), MYCN amplification (HR 0.351, *p =* 0.000), and stage 4 without oral metronomic MT (HR 0.366, *p =* 0.001) were significant adverse prognostic factors. Other factors, such as age, sex, and metastatic sites, had no significant impact on EFS or OS (Table 4).

Of the 126 patients with tumor recurrence or progression, 40 abandoned further treatment, all of whom died. Of the 86 patients who received salvage treatments, 47 died of tumor progression, and 38 were still alive at the end of the study. The 2-year OS after recurrence was 52.1 ± 6.0%.

### 3.4. The Toxicity of Oral Metronomic MT

The overall tolerance of oral metronomic MT was acceptable. MT was administered on an outpatient basis, and there was no treatment-related death. The most common toxicities were grade 1–2 hematological toxicity, accounting for about 80%. Grade 3 hematotoxicity accounted for 9%, no grade 4 toxicity. Grade 1–2 non-hematologic toxicity included transaminase elevation, nausea, gastritis, and creatinine elevation (Table 5). Oral metronomic MT was discontinued after 1–3 months in 5 patients, for various reasons.

## 4. Discussion

This study was to probe a new approach to improve the outcomes of patient with high-risk NB inaccessible to ASCT or GD2 antibody therapy. Among the 217 high-risk NB patients, MYCN amplification was positive in 28.8% of in the detected patients. Patients with stage 4 accounted for 91.2%. The overall response rate (CR/VGPR/PR) was 85.2% to comprehensive treatment. None of the patients received the treatment of ASCT or anti-GD2 antibodies. The 3-year EFS and OS rates for the entire cohort were 36.3% and 63.1%. Surprisingly, patients with stage 4 receiving oral metronomic MT after achieving CR/VGPR/PR had better 3-year EFS and OS rates than those without oral metronomic MT (42.5% vs. 29.6%, and 71.1% vs. 59.4%); The patients with oral metronomic MT including stage 3/4S patients had 3-year EFS and OS rates of 42.7% and 72.1%. MYCN amplification and stage 4 disease were significant adverse prognostic factors.

It is widely known that the EFS rate of high-risk NB patients treated with ASCT is slightly higher than that of conventional chemotherapy or no further therapy [1,2,3,4]. A study from the Beijing Children Hospital in China reported that the 3-year EFS rates for high-risk NB patients who underwent ASCT (196 patients), or not (296 patients), were 43.7% and 36.7%, respectively (*p =* 0.010); the corresponding 3-year OS rates were 57.6% and 53.5%, respectively (*p =* 0.153) [20]. However, there are few studies on the impact of oral metronomic MT on the survival of high-risk NB patients without ASCT treatment. A randomized study from Germany comparing ASCT versus oral MT as consolidation therapy in patients with high-risk NB showed that ASCT had increased 3-year EFS compared with those allocated MT (47% vs. 31%) but did not have significantly increased 3-year overall survival. Moreover, the patients in the oral MT group only received oral cyclophosphamide at the dose of 150 mg/m2 per day on days 1 to 8 monthly for three consecutive months [2]. However, this therapy does not really meet the criteria of metronomic oral MT.

Another German retrospective study was conducted to compare the long-term survival outcomes of anti-GD2-antibody ch14.18 group with oral MT group (without ASCT) or no consolidation therapy group (without ASCT and oral MT). The EFS was better in the anti-GD2 antibody group than those in the oral MT group and in non-consolidation group (5-year EFS 50.5%, 34.1%, and 25.9%, respectively). Multivariable Cox regression analysis revealed anti-GD2 antibody ch14.18 consolidation improved outcome compared to no consolidation. However, no difference exists between the MT and anti-GD2-treated patients. Further, although anti-GD2 treatment may have prevented late relapses in this study, oral MT also appeared to be effective at preventing relapses [21]. A retrospective non-randomized study analysis from Memorial Sloan Kettering Cancer Center questioned the importance of ASCT in the modern era. Among the patients with high-risk NB treated with anti-GD2 antibody 3F8/GM-CSF in 1st CR/VGPR, 60 had ASCT, and 110 had none prior to immunotherapy. Five-year EFS rate for ASCT versus non-ASCT patients was 65% versus 51% (*p =* 0.128), while OS rate was 76% versus 75% (*p =* 0.975), respectively. In multivariate analysis, ASCT was not prognostic. Despite the limitations of nonrandomized retrospective single institutional study, this sizable cohort did raise a hypothesis that ASCT may not be necessary for all patients, especially if there is an effective anti-GD2 immunotherapy for consolidation after induction therapy [22].

In our study, all high-risk NB patients did not undergo ASCT or anti-GD2 antibodies. Among them (cohort 2), 60% of patients with CR/VGPR/PR received oral metronomic MT, which led to 3-year EFS and OS rates of 42.7% and 72.1%. Patients in stage 4 receiving oral metronomic MT had a better survival rate than those who did not receive oral metronomic MT (cohort 1). These results were similar to those of ASCT from other studies but lower than those of anti-GD2 antibody immunotherapy [1,2,3,4,5,6,7,8,9,10]. The toxicities of oral metronomic MT were well tolerated, and cost was lower. The most common toxicities were grade 1–2 hematological toxicity, accounting for about 80%. Grade 3 hematological toxicities accounted for about 9%, no grade 4 toxicity. The starting time of oral metronomic MT was that peripheral white blood cells reached 3 × 10^9^/L after end of comprehensive therapy. Because all patients did not receive ASCT, bone marrow function recovered faster after the end of comprehensive treatment, and oral MT could be started quickly. Our study suggests that oral metronomic MT is an optimal option for high-risk NB patients who have achieved CR/VGPR/PR after comprehensive therapies without undergoing ASCT or anti-GD2 antibody therapy.

These metronomic MT drugs used in our study included low-dose cyclophosphamide, vinorelbine, topotecan/etoposide, and celecoxib as selective cyclooxygenase-2 inhibitors, which are potential anti-angiogenic agents [12]. Metronomic chemotherapy is defined as the continuous use of low dose chemotherapeutic agents. It was observed in vivo that tumors resistant to dose-intense therapy responded to the same agents at low continuous dosing [10]. Metronomic chemotherapy has been used in many refractory/relapsed solid tumors. Many studies have shown that low-dose metronomic chemotherapy combined with anti-angiogenic drugs can markedly enhance the anti-angiogenic efficacy [11,12,13,14,16,17]. Zapletalova reported the results of a multicenter study on metronomic chemotherapy with the Combined Oral Metronomic Biodifferentiating Antiangiogenic Treatment (COMBAT) regimen in advanced pediatric malignancies. The COMBAT regimen includes low-dose daily temozolomide, etoposide, celecoxib, vitamin D, fenofibrate, and retinoic acid. They found that COMBAT was a feasible and effective treatment option for patients with relapsed/refractory malignancies, and it was well tolerated with low toxicity. They proposed introducing metronomic chemotherapy at earlier phases of antitumor treatment to allow further exploitation of the potential of low-dose metronomic chemotherapy, for instance, as MT [11]. Kieran et al. demonstrated the feasibility of combining low-dose metronomic chemotherapy (alternating cyclophosphamide and etoposide) and anti-angiogenic agents (celecoxib and thalidomide) in children with recurrent or progressive solid tumors. Their results showed that regimen was well tolerated and had some activity in some tumors [23]. These studies suggest that low-dose metronomic chemotherapy combined with anti-angiogenic drugs as MT may be beneficial to patients with high-risk solid tumors. Recent result from a randomized clinical trial has demonstrated low-dose metronomic MT can improve outcomes in patients with pediatric high-risk rhabdomyosarcoma (RMS) [17]. Our study also showed low-dose oral metronomic MT can benefit high-risk NB patients without ASCT or anti-GD2 antibodies therapy.

Based on the Children’s Oncology Group (COG) neuroblastoma high-risk group assignment criteria, high-risk NB is defined as stage 4 disease with age ≥ 18 months; stage 2, 3, and 4S disease with MYCN amplification; and stage 3 disease with age >18 months with unfavorable pathology without MYCN amplification. High-risk NB patients receive the same treatment regimen and have a poor prognosis. As far as we know, MYCN amplification strongly predicts a poor prognosis in NB. A study from South Korea reported that there was no survival difference between patients with and without MYCN amplified in high-risk NB [24]. But, inferior EFS and OS in high-risk NB patients with MYCN amplification have been reported from the COG and other studies [25,26,27,28]. One study from Japan showed MYCN amplification was the most favorable prognostic factor for EFS of high-risk NB [29]. Our results showed that the patients with MYCN amplified had worse the EFS and OS rates than those without MYCN amplified in high-risk NB. For stage 4 NB patients with MYCN amplification, new treatments need to be explored.

Another prognostic factor is staging. Most high-risk NB patients are in the stage 4, but the stage 3 patients with MYCN amplification or unfavorable pathology are also defined as high-risk patients. In our study, the patients with stage 4 accounted for 91.2%, while stage 3 patients with MYCN amplification or unfavorable pathology accounted for only 8.3%, and the EFS and OS rates in the stage 4 patients were worse than those in the stage 3 patients with MYCN amplification or unfavorable pathology (33.8% vs. 62.2%; 60.3% vs. 83.3%). In addition, stage 4 patients with CR/VGPR/PR who did not receive oral metronomic MT had worse survival than those with oral metronomic MT. Univariate and multivariate analysis also confirmed these results. Thus, our study demonstrated that stage 4 disease and MYCN amplification were independent adverse prognostic factors for high-risk NB patients. These results are similar to those reported in other studies [25,26,27,28,30].

Recurrence in patients initially defined as high-risk NB have a very poor prognosis, with an overall survival of only 11.0 months, and the reported 5-year overall survival rate for children after the initial relapse of NB is 6–20% [31,32,33]. The OS rate is closely related to whether the patients receive salvage treatment after recurrence. In this study, 126 patients experienced tumor recurrence or progression, and 40 of them abandoned further treatment and died. The remaining 86 patients received active salvage chemotherapy, surgery, radiotherapy, or other treatment, and most of these patients had effective tumor control after salvage therapy. Some patients relapsed repeatedly and were treated repeatedly. The 2-year overall survival rate after recurrence was 52.1%, and some patients survived for more than 5 years after recurrence. This may explain the reason of the high 3-year overall survival rate as 60–70% in our study. These relapsed patients who had not previously received ASCT tolerated salvage chemotherapy well. Our salvage chemotherapy regimen included some drugs that were not used as initial treatment and were effective in patients with recurrent NB. Therefore, our results suggest that the survival of high-risk NB patients with recurrence can be prolonged by salvage treatment.

## 5. Conclusions

Our study suggests that oral metronomic MT is an optimal option for high-risk NB patients without ASCT or anti-GD2 antibodies therapy. Stage 4, *MYCN* amplification, and stage 4 patients not receiving oral metronomic MT after CR/VGPR/PR were independent adverse prognostic factors for high-risk NB patients without ASCT or anti-GD2 antibodies therapy. Additionally, high-risk NB patients with recurrence can obtain a survival benefit from salvage therapy.

## Figures and Tables

**Figure 1 cancers-13-03494-f001:**
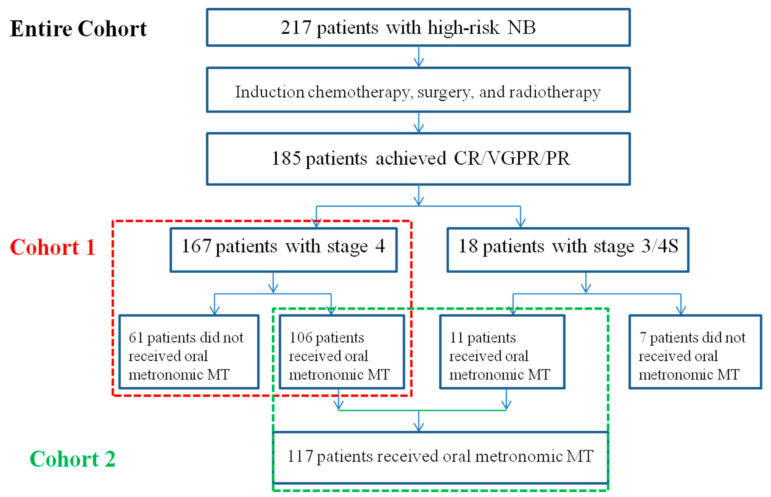
Therapy flow chart and cohorts of the patients with high-risk NB. NB: neuroblastoma; CR: complete response; VGPR: very good partial response; PR: partial response. MT: maintenance therapy.

**Figure 2 cancers-13-03494-f002:**
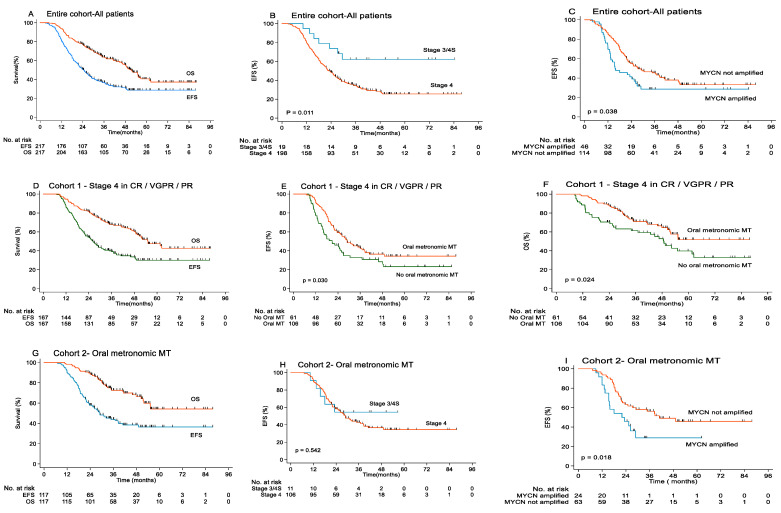
Survival curves of high-risk NB not treated with ASCT or anti-GD2 antibodies. (**A**) EFS and OS survival of all patients in the entire Cohort; (**B**) EFS of patients with different stage in the entire Cohort; (**C**) EFS of patients with MYCN amplified or no amplified in the entire Cohort; (**D**) EFS and OS survival in Cohort 1; (**E**) EFS of patients with or without metronomic MT in Cohort 1; (**F**) OS of patients with or without metronomic MT in Cohort 1; (**G**) EFS and OS survival in Cohort 2; (**H**) EFS of patients with different stage in Cohort 2; (**I**) EFS of patients with MYCN amplified or no amplified in Cohort 2.

**Table 1 cancers-13-03494-t001:** Chemotherapy regimen for newly diagnosed high-risk NB.

Chemotherapy Regimens	Drugs Dosage and Administration
Induction Chemotherapy:	
CAV (Cycle 1, 3, 5, and 7)	Cyclophosphamide (CTX) 1 g/m^2^, iv drip for 0.5 h, d1–2 Mesna 330 mg/m^2^ 0, 4, and 8 h after CTX, iv, d1–2 Vincristine 1.5 mg/m^2^, iv, d1 Pirarubicin 50 mg/m^2^, iv, d1
VIP (Cycle 2, 4, 6, and 8)	Cisplatin 25 mg/m^2^, iv drip for 3 h, d1–4 Etoposide 100 mg/m^2^, iv drip for 3 h, d1–4 Ifosfamide (IFO) 1.5 g/m^2^, iv drip for 3 h, d1–4 Mesna 300 mg/m^2^, iv 0, 4, 8 h after IFO, d1–4
Second-line chemotherapy:	
VIT	Vincristine 1.5 mg/m^2^, iv, d1 Irinotecan 50 mg/m^2^, iv drip for 1.5 h, d1–5 Temozolomide 100 mg/m^2^, po, d1–5
Maintenance therapy:	
Oral metronomic anti-angiogenic agents	At months 1, 3, 5, 7, 9, and 11: Cyclophosphamide 25–50 mg/m^2^, po, d1–30 Vinorelbine 40 mg/m^2^, po, qw × 3 Etoposide 25 mg/m^2^, po, d1–21 Celecoxib 200 mg/m^2^, po, bid d1–30 At months 2, 4, 6, 8, 10, and 12: Cyclophosphamide 25-50 mg/m^2^, po, d1–30 Vinorelbine 40 mg/m^2^, po, qw × 3 Topotecan ^a^ 1.4 mg/m^2^, po, d1–5 Celecoxib 200 mg/m^2^, po, bid d1–30

^a^: Since December 2017, topotecan has been discontinued due to no oral topotecan supply; NB: neuroblastoma; CAV: cyclophosphamide, pirarubicin, and vincristine; iv: intravenous injection; VIP: etoposide, ifosfamide, and cisplatin; po: oral administration; VIT: vincristine, irinotecan, and temozolomide.

**Table 2 cancers-13-03494-t002:** Clinical features of 217 high-risk NB patients without ASCT or anti-GD2 antibody therapy.

Feature	Entire Cohort (*n* = 217)	Cohort 1 (*n* = 167)	Cohort 2 (*n* = 117)
Sex			
Male	145 (66.8%)	122 (65.9%)	76 (65%)
Female	72 (33.2%)	63 (34.1%)	41 (35%)
Age			
Age ≥ 18 months at diagnosis	200 (92.2%)	171 (92.4%)	107 (91.5%)
Age < 18 months at diagnosis	17 (7.8%)	14 (7.6%)	10 (8.5%)
Stage			
INSS Stage 4	198 (91.2%)	167 (90.3%)	106 (90.6%)
INSS Stage 3	18 (8.3%)		10 (8.5%)
INSS Stage 4S	1 (0.5%)		1 (0.9%)
Primary tumor site			
Adrenal gland primary	121 (55.8%)	106 (57.3%)	72 (61.5%)
Retroperitoneal primary	71 (32.7%)	59 (31.9%)	36 (30.8%)
Mediastinum primary	17 (7.8%)	12 (6.5%)	5 (4.3%)
Other primary sites	8 (3.7%)	8 (4.3%)	4 (3.4%)
Metastatic sites			
Bone marrow/bone metastasis	172 (79.3%)	146 (78.9%)	94 (80.3%)
Other metastasis sites	45 (20.7%)	39 (21.1%)	23 (19.7%)
MYCN status			
MYCN amplified	46 (28.8%)	39 (28.3%)	24 (27.6%)
MYCN not amplified	114 (71.3%)	99 (53.5%)	63 (72.4%)
MYCN unknown	57 (26.3%)	47 (25.4%)	30 (25.6%)

NB: neuroblastoma. ASCT: autologous stem cell transplantation; INSS: International Neuroblastoma Staging System; Cohort 1: Stage 4 Patients with CR/VGPR/PR received or did not receive oral metronomic MT; Cohort 2: Patients with CR/VGPR/PR received oral metronomic MT.

**Table 3 cancers-13-03494-t003:** Treatment outcome of high-risk NB patients not treated with ASCT or anti-GD2 antibody.

	*N*	3-Year EFS (%)	*p*-Value	3-Year OS (%)	*p*-Value
Entire Cohort	217	36.3 ± 3.4		63.1 ± 3.4	
Age					
Age ≥ 18 months	200	35.5 ± 3.5	0.63	63.6 ± 3.5	0.575
Age < 18 months	17	45.3 ± 12.4		57.4 ± 12.3	
Sex					
Male	145	36.4 ± 4.1%	0.752	64.1 ± 4.1	0.965
Female	72	36.0 ± 6.0%		61.1 ± 6.1	
Stage					
INSS Stage 4	198	33.8 ± 3.5	0.011	60.3 ± 3.6	0.008
INSS Stage 3/4S	19	62.2 ± 11.4		83.3 ± 11.2	
MYCN status					
MYCN amplified	46	28.6 ± 7.0	0.038	48.6 ± 8.0	0.008
MYCN no amplified	114	44.3 ± 4.8		74.0 ± 4.3	
Metastatic sites					
BM/bone metastasis	172	32.6 ± 3.7	0.102	60.4 ± 3.9	0.097
Other metastasis sites	45	47.0 ± 7.7		70.5 ± 7.3	
Cohort 1 ^a^	167	37.8 ± 3.9		66.9 ± 3.8	
Age					
Age ≥ 18 months	154	36.0 ± 0.4	0.212	67.5 ± 3.9	0.910
Age < 18 months	13	59.8 ± 14		59.2 ± 14.1	
Sex					
Male	110	37.2 ± 4.8	0.953	68.2 ± 4.6	0.652
Female	57	38.8 ± 6.9		64.2 ± 4.7	
MYCN status					
MYCN amplified	27	12.3 ± 9.9	0.019	37.8 ± 10.9	0.000
MYCN not amplified	93	46.1 ± 5.3		76.7 ± 4.6	
Oral MT					
Oral metronomic MT	106	42.5 ± 5.1	0.017	71.1 ± 6.7	0.022
No oral metronomic MT	61	29.6 ± 6		59.4 ± 6.4	
Cohort 2 ^b^	117	42.7 ± 4.8		72.1 ± 4.5	
Age					
Age ≥ 18 months	107	41.4 ± 5.0	0.261	72.2 ± 4.6	0.563
Age < 18 months	10	68.6 ± 15.1		77.8 ± 13.9	
Sex					
Male	76	46.8 ± 5.9	0.513	70.5 ± 5.5	0.600
Female	41	36.2 ± 8.3		76.6 ± 7.4	
MYCN status					
MYCN amplified	24	16.8 ± 13	0.026	56.3 ± 13.4	0.042
MYCN not amplified	63	53.9 ± 6.5		78.1 ± 5.7	
Stage					
Stage 4	106	42.5 ± 5.1	0.556	71.1 ± 4.7	0.167
Stage 3/4S	11	54.5 ± 15		88.9 ± 10.5	

Abbreviations: NB: neuroblastoma; ASCT: autologous stem cell transplantation; BM: bone marrow; EFS: event free survival; OS: overall survival; ^a^: Cohort 1 included patients in stage 4 with CR/VGPR/PR received or did not receive oral metronomic MT; ^b^: Cohort 2 included patients with CR/VGPR/PR received oral metronomic MT.

**Table 4 cancers-13-03494-t004:** Univariate and multivariate analyses of prognostic factors of high-risk NB patients ^a^.

Category	No	3-Year Event-Free Survival			3-Year Overall Survival		
		Univariate	Multivariate ^b^		Univariate	Multivariate ^b^	
		*p-*Value	HR (95% CI)	*p-*Value	*p-*Value	HR (95% CI)	*p-*Value
Age							
≥18 months	200	0.631	0.680	0.285	0.576	1.075	0.859
<18 months	17		(0.335, 1.379)			(0.481, 2.402)	
Sex							
Male	145	0.752	1. 218	0.362	0.965	1.125	0.649
Female	72		(0.798, 1.858)			(0.677, 1.869)	
Stage							
Stage 4	198	0.014	0.246	0.002	0.014	0.147	0.003
Stage 3/4S	19		(0.102, 0.594)			(0.041, 0.525)	
*MYCN* amplified							
Yes	46	0.040	0.462	0.001	0.009	0.351	0.000
No	114		(0.292, 0.732)			(0.209, 0.590)	
Metastasis sites							
BM/Bone	172	0.104	1.151	0.608	0.100	1.026	0.937
Other	45		(0.672, 1.972)			(0.550, 1.914)	
Stage 4 in CR/VGPR/PR							
Oral MT ^c^	106	0.030	0.559	0.014	0.024	0.366	0.001
No oral MT	61		(0.351, 0.891)			(0.197, 0.680)	

Abbreviations: BM: bone marrow; ^a^: High-risk NB patients not treated with ASCT or anti-GD2 antibody; ^b^: Multivariate analysis was adjusted for age, sex, stage, metastasis sites, MYCN amplified and stage 4 with oral metronomic MT in CR/VGPR/PR; ^c^: Oral metronomic agents maintenance therapy.

**Table 5 cancers-13-03494-t005:** Toxicity of oral metronomic drugs maintenance therapy.

	Toxicity Grading [*n* (%)]				
	0	1	2	3	4
Hemoglobin	27 (23.1)	43 (36.8)	45 (38.5)	2 (1.7)	0 (0)
White blood cell	13 (11.1)	46 (39.3)	49 (41.9)	9 (7.7)	0 (0)
Platelets	116 (99.1)	1 (0.9)	0 (0)	0 (0)	0 (0)
Transaminase	104 (88.9)	10 (8.5)	2 (1.7)	1 (0.9)	0 (0)
Nausea	114 (97.4)	1 (0.9)	2 (1.7)	0 (0)	0 (0)
Creatinine	113 (96.6)	4 (3.4)	0 (0)	0 (0)	0 (0)
Gastritis	114 (97.4)	1 (0.9)	2 (1.7)	0 (0)	0 (0)

## Data Availability

The data that support the findings of this study are openly available in Research Data Deposit at https://www.researchdata.org.cn, reference number RDDA2021002014 (accessed on 3 June 2021).

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
