# Peer review of "Oral Metronomic Maintenance Therapy Can Improve Survival in High-Risk Neuroblastoma Patients Not Treated with ASCT or Anti-GD2 Antibodies"

_cancers, 2021, doi:10.3390/cancers13143494_

Round 1
Reviewer 1 Report
The authors report that oral maintenance chemotherapy can improve outcome for high-risk neuroblastoma patients who have no access to or do not want receive high-dose chemotherapy and/or anti-GD2 antibody treatment. This is an important contribution to the scientific community and a base to reconsider the clinical practice in countries with less economical resources, but also in other countries for patients under special medical conditions. The manuscript is well written.
Although not randomized, a comparison with the outcome (EFS, OS) of the patients treated at the same time with high-dose chemotherapy and/or anti-GD2 immunotherapy would be more convincing than the given references in the literature. However, these data may be not available at this time.
Points of criticism:
Line 35
Please include the number of patients with or without oral metronomic chemotherapy.
Line 63 and 434-450
For several papers updates are available such as
- 2: update in Br J Cancer. 2018 Aug;119(3):282-290. doi: 10.1038/s41416-018-0169-8. Epub 2018 Jul 11.PMID: 29991700
- 5: update in Cochrane Database Syst Rev. 2015 Oct 5;(10):CD006301. doi: 10.1002/14651858.CD006301.pub4. PMID: 26436598
- 6: update in Clin Cancer Res 2021 Apr 15;27(8):2179-2189. doi: 10.1158/1078-0432.CCR-20-3909. PMID: 33504555
This may modify the statement of ‘no OS benefit’ (line 35) and the range of the long-term survival rates (line 69).
Line 102
Include as exclusion criterion: no high-dose chemotherapy (ASCT) and/or no anti-GD2 immunotherapy.
Lines 99 and 106
The reference is missing for ‘unfavorable histology’ and ‘pathology classification system’
Line 121
Did the patients without oral metronomic chemotherapy receive any other treatment instead? Please clarify. This is nowhere stated throughout the manuscript and important to know.
Line 123, Table 1
Include the recommended time periods for the infusions (drips)
Line 159, Table 2
Cohort 1 (stage 4 with and without MT): N=167 is correct, but all the subcohorts below include stage 3 and 4S and add to N=185. Omit the stage 3/4S patients and correct the figures accordingly.
Substitute ‘MYCN no amplified’ by ‘MYCN not amplified’ throughout the manuscript.
Line 177, Figure 1
Substitute GPR by VGPR (very good partial response) as stated elsewhere
Line169/170 vs. Figure 2 A-C
’18 had PD after comprehensive therapies’. Does that mean that no patient had PD before the end of induction therapy or were excluded from survival analysis? Usually, the Kaplan-Meier curves drop right from the start of induction chemotherapy (EFS>OS). The panels 2A, 2B and 2C are too tiny to indicate where the count starts: at diagnosis or after the end of induction therapy around 6 months. Please, clarify.
Line 280
Only grade 1 and 2 hematological toxicities (CTCAE) appear surprisingly low after intensive induction chemotherapy -even if the metronomic chemotherapy itself is low toxic. Any explanation (discussion section)?
Line 294
Correct the 29.4% by 29.6% as shown in Table 3 line 224.
Line 295
Add after MT ‘including stage 3/4S patients’ for clarification
Line 330
Add ‘(cohort 2)’
Line 331
Add ‘s’ to patient
Lines 390-405 and 410-412
Omit the paragraph and the last sentence. This is not the focus of the paper.
Reviewer 2 Report
Authors present a significant article on the reality of treating High-risk Nb in a huge country with very significant children's population and from where no relevant information is available to the research community. Authors present honestly the reality of the families in China and the treatment availability. From this, they devised a strong treatment program with spoecific aims and more importantly with feasible means for their population. Results are significant and reveal important messages for the NB field in general.
The large patient size is notable with >200 pts treated in 5 years with limited diagnostic tools (MIBG is unavailable) but systematic approach is clearly described. The metronomic concept of maintenance therapy in the absence of immunotherapy has not been explored before in such a large population. The results are notable with EFS/OS clearly superior to no treatment and reminder of 1990s survival outcomes reported in Western countries. This results are important for current baseline comparison intends of future studies and to dissect the relevance of ASCT or anti-GD2 immunotherapy in the context of poost induction managment of patients showing objective responses.
Minor comments:
- Figure 1 is confusing. Numbers do not add up
- It would be very important to specifically present survival data on patients having achieved CR. Those patients also might have some differences in the predictive markers of survival (MYCN for instance).
Reviewer 3 Report
The authors described the survival outcome of a cohort of high risk neuroblastoma (NB) patients treated with or without metronomic, oral maintenance chemotherapy (MT) in a setting in which high-dose chemotherapy with stem cell transplantation and anti-GD2 antibody were not available. The author found that oral MT may benefit patients with stage 4 high-risk NB (Cohort 1).
A major shortcoming is the vague description of study design. How were patients chosen to receive oral MT? Was there a randomization? Were there potential causes of selection bias (e.g. by physician choice, who might prefer patients with worse response to receive oral MT)? Was other treatment allowed in the non-oral MT group?
Furthermore, calling the two subgroup analyses as Cohort 1 and Cohort 2, which is confusing because the two "cohorts" have overlapping patients. If the authors wish to identify NB patient subpopulations who are most likely benefit from oral MT, they should do two things:
(1) construct a multivariate Cox model of all patients who had achieved CR/VGPR/PR and had a choice of oral MT and analyze the impact of each variable, especially age, stage, MYCN status and oral MT status;
(2) compare the survival between patients with and without oral MT in each of the subgroups, i.e. infants, non-infant stage 4, non-infant stage 3/4, MYCN amplified, MYCN non-amplified
Other comments:
Case No. of cohort 2 (oral MT) = 167 stage 4 + 11 other HR patients.
Who are the "other" high-risk patients who are not stage 4? Were they all stage 3/4S?
Since 185 - 167 = 18, where are the other 7 patients? Did they receive oral MT? Were they excluded from analysis?
Please revise this sentence in Abstract:
"Stage 4 disease, MYCN amplification and stage 4 in CR/VGPR/PR not receiving oral metronomic MT..."
It was difficult to understand.
Figure 1 was difficult to understand. I suggest to convert it into a standard CONSORT diagram. If there is a randomization or comparison of patient flow, please specify. Please also describe the patient flow of non-stage 4 patients (n = 185 - 167 = 18).
Table 1: VIT was listed as 2nd line treatment. How many patients have used this regimen? What is the criteria of using VIT? Did patients who did not received MT had received VIT as maintenance?
Table 2 and Table 3 need to be reorganized to list each variable more clearly (e.g. INSS Stage 4/3/4S, with each slash representing a new row with indentation).
In Table 3, Please also compare the different age groups and MYCN status within Cohort 1.
Table 4. The case number of the last three rows (Stage 4) was different from the other rows. There seems to be at least two different models in this table. Why did the authors analyze only stage 4 patients? If the other high-risk patients were included, was the result/trend the same? The authors would better use the cohort before randomization or oral MT treatment choice (n = 185) to construct the multivariate Cox model and to discuss whether oral MT is an independent predictor of better survival.
Discussion:
Please compare with other studies of maintenance therapy e.g. the Etoposide regimen used in Germany.
Round 2
Reviewer 3 Report
The authors have revised the manuscript according to previous comments and suggestions. The results are sound and serve as a good roadmap to settings where ASCT and anti-GD2 are not routinely available and to the design of future protocols.
A very minor suggestion is to add in a SPACE on Page 4, Table 1, Line 10 (Cyclophosphamide), between the first comma (,) and "po".
Author Response
Thank you for your careful review.
A SPACE has been added on Page 4, Table 1, Line 10 (Cyclophosphamide), between the first comma (,) and "po".